# Modeling Field Electron Emission from a Flat Au (100) Surface with Density-Functional Theory

**Yiming Li \***, **Joshua Mann**  **and James Rosenzweig**

Department of Physics & Astronomy, University of California Los Angeles, 475 Portola Plaza, Los Angeles, CA 90095, USA; jomann@physics.ucla.edu (J.M.); rosen@physics.ucla.edu (J.R.)
\* Correspondence: yimingli20@ucla.edu

**Abstract:** Field electron emission, or electron tunneling through a potential energy (PE) barrier under the influence of a strong electrostatic (ES) or radio frequency (RF) field, is of broad interest to the accelerator physics community. For example, it is the source of undesirable dark currents in resonant cavities, providing a limit to high-field operation. Field electron emission can also be applied to quasi-statically model electron emission induced by the electric field in a laser pulse. The classical approach to field electron emission is the Fowler–Nordheim (FN) framework, which incorporates a simplified PE profile and various assumptions. Here, we build a more realistic model using the PE and charge densities derived from a density-functional theory (DFT) calculation. We examine the correction factors associated with each model assumption. Compared to the FN framework, our results can be extended up to 80 GV/m, a limit that has been reached in laser-induced strong field emission scenarios.

**Keywords:** field electron emission; DFT; Au (100); Fowler–Nordheim; laser field emission

## 1. Introduction

Field electron emission from metals is due to electron tunneling from a metallic surface under a strong ES or RF field. This is to be differentiated from laser-induced field emission, which is an electron emission induced by the electric field in a laser pulse. Field emission induces surface breakdown in RF cavities with surface fields of 300 to 400 MV/m, which, after geometric enhancement, yields local surface fields up to 10 GV/m [1]. Surface breakdown is the source of undesirable dark currents from high-gradient accelerating structures, which dissipate power from the accelerating RF field. Dark current also gives rise to significant pollution of the experimental environment downstream of the electron source. Modeling such dark currents, however, requires an accurate understanding of the physical and chemical defects of the emitting surface. Oftentimes, these defects amount to an effective enhancement of the field. We may then accurately model these emissions via enhanced fields applied to one-dimensional, flat, chemically pure surfaces. This is a major focus of the present paper.

Furthermore, peak surface-enhanced electric fields typically exceed 10 GV/m in laser-induced field emission experiments. Peak transient fields as large as 80 GV/m have been achieved on gold-coated nanoblades with an ultrafast, 800 nm laser via plasmonic enhancement [2,3]. For these parameters, the Keldysh parameter is valued at $\gamma \approx 0.23$ [4], identifying the tunneling process as dominant compared to the multiphoton emission process. Additionally, the transverse motion induced by these fields is negligible for such a nanostructure—see Appendix A. It is therefore appropriate to model laser-induced field emission for this case as a quasi-static one-dimensional field emitter, indicating a need to explore ES field emission at this extreme.

The focus of this paper is field electron emission, so the word "field" will refer to the surface-enhanced ES field at a metallic emitter in the following discussion. By convention,

this field is directed into the emitter such that electrons in a vacuum are extracted from the surface. Note that this paper attempts to model field electron emission for fields up to 80 GV/m. Only the lower limit (usually less than 10 GV/m) is directly applicable to field electron emission experiments (with ES or RF fields), while the upper limit may be used quasi-statically to model laser-induced field emission.

The conventional model of field electron emission is the Fowler–Nordheim (FN) framework. It is a one-dimensional quantum mechanical model that arises from solving the time-independent Schrödinger equation (TISE). The FN framework typically employs two simplified PEs. One is the exact triangular (ET) PE of the form $U(z) = \phi - eFz$ outside the surface, where $\phi$ is the work function, $e$ the elementary charge, $F$ the ES field, and $z$ the distance from the surface. It is set to a constant within the metal to reflect the effectively free electron dynamics. Although the ET PE is an unrealistic model of a metallic surface, it admits analytical solutions under the FN framework [5]. The other model of interest is the Schottky–Nordheim (SN) PE given by $U(z) = \phi - eFz - \frac{e^2}{16\pi\epsilon_0 z}$, where the last term corresponds to the image-charge PE [6]. The image-charge PE oversimplifies the electron exchange and correlation effects, but it has been widely used in the field electron emission literature. In this study, we build more realistic one-dimensional PE profiles based on DFT calculations. We carefully analyze the output of these calculations to build physically meaningful PE profiles as a function of the field.

Instead of solving the TISE exactly, the conventional FN framework uses the Wentzel–Kramers–Brillouin (WKB) approximation to find the electron transmission coefficient (the probability that an incident electron would transmit through a PE barrier). The transmission coefficient as a function of the electron normal energy is given by $D(E_n) = e^{-G(E_n)}$, where the Gamow coefficient is $G(E_n) = \frac{\sqrt{8m_e}}{\hbar} \int \sqrt{U(z) - E_n}\, dz$, with definite integration performed between the two classical turning points $z_{lim}$ provided by $U(z_{lim}) = E_n$. The variable $m_e$ is the electron mass, $U(z)$ is the electron PE as a function of longitudinal position, and $E_n$ is the normal energy (total energy minus the surface-parallel kinetic energy) of the electronic state under consideration. It has been noted, however, that the WKB approximation is only strictly true at low fields and for smooth PE profiles [7]. High-precision numerical techniques have therefore been developed to calculate the transmission coefficient. For the ET PE, there are exact solutions in terms of Airy functions and their derivatives [8]. No closed-form solution has been found for the SN PE, not to mention the PE we derive from DFT calculations. Mayer has used the transfer-matrix (TM) method to numerically solve the transmission coefficients from ET and SN PE. The TM method discretizes the PE and takes it to be a piecewise constant, allowing us to solve for the wavefunction by applying boundary conditions within each interval [9]. In this study, we compare the TM method with the Runge–Kutta method for obtaining the transmission coefficient.

In the FN framework, the decay width, describing the exponential dependence of the transmission coefficient on normal energy, is defined as $d^{-1} = -\frac{\partial G}{\partial E_n}$. A first-order Taylor expansion of $G$ around the Fermi level leads to a simple formula for the emission current density $J = z_s d_F^2 D_F$, where $z_s$ is the Sommerfeld's free electron supply constant, and the subscript $F$ indicates evaluation at the Fermi level [5]. In this study, we substitute the first-order Taylor expansion by an integral in the energy space and examine the correction factor to the linear approximation. The emission current calculated following our numerical procedure is more accurate, especially when used quasi-statically to model laser-induced field emission.

## 2. Materials and Methods

### 2.1. DFT Calculation

There are ongoing experimental and computational efforts to study laser field emission from Au-coated nanoblades, which are one-dimensional structures with a sharp edge used for plasmonic field enhancement [3,10]. Inspired by such efforts, we chose to study a face-centered cubic Au lattice with (100) surface orientation. We used the package

JDFTx to implement all DFT calculations [11]. As shown in Figure 1, each Au (100) slab consisted of 15 layers of Au atoms, with each layer extending infinitely on a (100) lattice plane with periodic boundary conditions. The innermost 9 layers were fixed, while the outermost 3 layers on either side were optimized along the direction perpendicular to the surface to minimize the Gibbs free energy. JDFTx uses periodic boundary conditions in all directions, so an infinite array of Au (100) slabs was modeled. The periodic slab-to-slab spacing was equivalent to 26 layers such that the interaction between neighboring slabs was negligible [12]. The lattice constant of Au was fixed to 771 pm [13]. We considered both the ultrasoft GBRV pseudopotential (PP) [14] and the norm-conserving SG15 PP [15]. Although a norm-conserving PP generally provides more accurate results at the cost of computational efficiency, our calculation using the SG15 PP failed to converge. We therefore chose to continue with the GBRV PP. For electron exchange-correlation (XC) functionals, previous studies of similar systems have used the generalized gradient approximation (GGA) of the form Perdew–Wang 1991 (PW91) and Perdew–Burke–Ernzerhof (PBE) [12,16–18]. It turned out that the PBE XC functional resulted in a more reasonable Kohn–Sham (KS) PE profile (whose creation is described in Section 2.2) for our system, whereas the PW91 profile had a jump discontinuity and shot above the vacuum level before converging back (Figure 2). We therefore chose the PBE XC functional for all the following calculations. Our DFT calculation also used truncated Coulomb potentials [19], total energy minimization with auxiliary Hamiltonian [20], and smooth electrostatic potentials with atom-potential subtraction [21]. We performed a convergence test on the $12 \times 12 \times 1$, $16 \times 16 \times 1$, and $20 \times 20 \times 1$ k-point grids, and we chose $20 \times 20 \times 1$ for best convergence. Since the wavefunctions were expanded with a planewave basis, we performed an additional convergence test on the planewave energy cutoff, with the optimal value at 50 Hartrees (the atomic unit of energy, 1 a.u. = 1 Hartree $\approx$ 27.2114 eV).

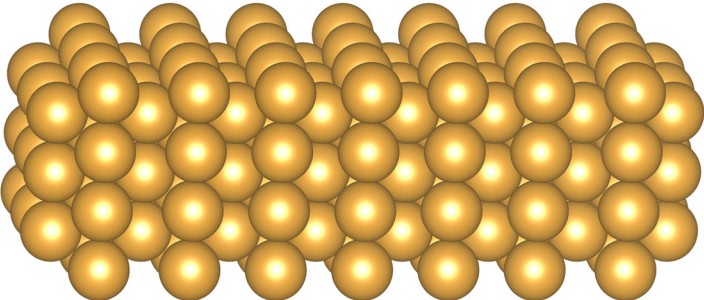

**Figure 1.** The Au (100) slab used in DFT calculations.

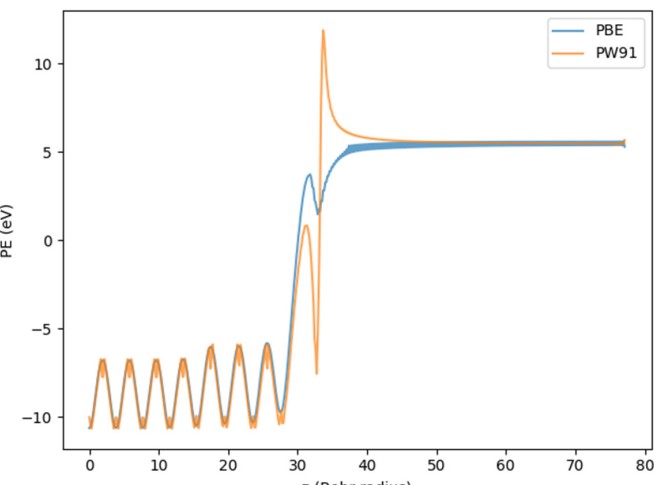

**Figure 2.** One-dimensional KS PE with a suboptimal k-point grid and a smaller planewave energy cutoff for the purpose of XC selection, using the PBE and PW91 XC functionals. The PE profiles are

plotted along the normal direction through the surface, from the center of the slab to the center of the vacuum separating neighboring slabs. Periodic oscillations below $z \approx 30$ Bohr radii (the atomic unit of length, 1 a.u. = 1 Bohr radius $\approx 52.92$ pm) correspond to each layer of atoms inside the slab. Note that the PBE profile can be optimized by increasing the size of k-point grid and planewave energy cutoff and applying a Gaussian filter.

## 2.2. DFT Results Post-Processing

JDFTx yielded the KS PE on a three-dimensional grid in real space. To construct a one-dimensional model, we extracted the PE along the longitudinal line $x = a/2$, $y = 0$, which avoided all nuclei [22]. To decrease the high-frequency noise in the KS PE outside the slab, we employed a Gaussian filter with a standard deviation of one grid-point separation. To obtain the KS PE at nonzero field, one solution was to implement a new DFT calculation with such a field. However, the KS PE failed to converge to the vacuum level for fields greater than 1 GV/m, a problem also previously encountered in the literature [22]. Instead, we used the KS PE from the zero-field calculation and added a linear PE in vacuum under the dipole approximation. The location where the field begins is known as the electrical surface, which can be calculated from the electrical centroid rule [23,24]. It states that the electrical surface coincides with the centroid of the induced charge densities or the excess charge densities induced by an applied field. As shown in Figure 3, an applied field induced an excess of electrons on the rightmost surface of the slab, as shown by the maximum-induced electron density (the opposite of induced charge density). Note that the centroid, or the electrical surface, did not coincide with the geometric surface of the slab. The centroid was calculated for fields 100 MV/m, 300 MV/m, 700 MV/m, and 1 GV/m, using the charge densities from corresponding DFT calculations and from the zero-field DFT calculation. As the fluctuations in the centroid were small, we assumed that the electrical surface did not change significantly for fields greater than 1 GV/m. We therefore took the average of the centroid locations to be the electrical surface.

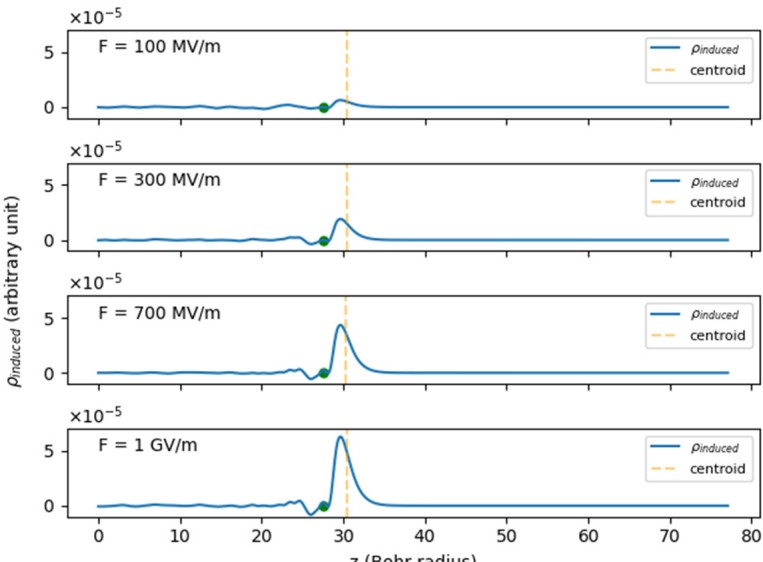

**Figure 3.** Longitudinal profiles of the induced electron density $\rho_{induced}$ at fields of 100 MV/m, 300 MV/m, 700 MV/m, and 1 GV/m. The center of the slab is at 0 Bohr radii, and the solid green dot represents the position of the outermost atom, which is fixed for all the fields presented. The dashed orange lines show the centroid locations of the induced electron density, which are 30.40 Bohr radii (F = 100 MV/m), 30.49 Bohr radii (F = 300 MV/m), 30.34 Bohr radii (F = 700 MV/m), and 30.40 Bohr radii (F = 1 GV/m).

The zero-field KS PE is presented along with the ET and SN PEs (Figure 4). The periodic oscillations in the KS PE, up to $z \approx 30$ Bohr radii, were due to the nuclei inside the slab. Regions beyond $z \approx 30$ Bohr radii showed a gradual transition to vacuum. The ET and SN PEs inside the slab were set to the average of the KS PE at $-8.88$ eV. The Fermi level was placed at 0 eV, so the vacuum level of the PE was equal to the work function of Au (100) at 5.22 eV [13]. The depth of the PE well was therefore 14.10 eV. The jump discontinuity in the ET PE and the image plane of the SN PE both coincided with the electrical surface. Note also that the KS PE had a "bump" around $z \approx 34$ Bohr radii, where we expected the curve to be smooth. It was therefore reasonable to generate a piecewise-defined PE to remove this feature. The first piece we chose to be the same as the KS PE up to some point before the observed bump, while the second piece should have been smooth and monotonic while ultimately approaching the vacuum level. A simple model for the second piece was the image-charge PE. The problem at hand was to determine the optimal location that connected these two pieces. We wanted the new PE to be continuous and differentiable, resulting in two boundary conditions that completely specify the location of this junction. We termed this piecewise-defined PE the image-charge corrected (IC) PE. Finally, under an applied field, these four PEs had different barrier heights: the ET PE was the highest, followed by the KS, IC, and the SN PEs. Generally, the higher the PE barrier, the smaller the emission currents tended to be, as anticipated by the WKB approximation.

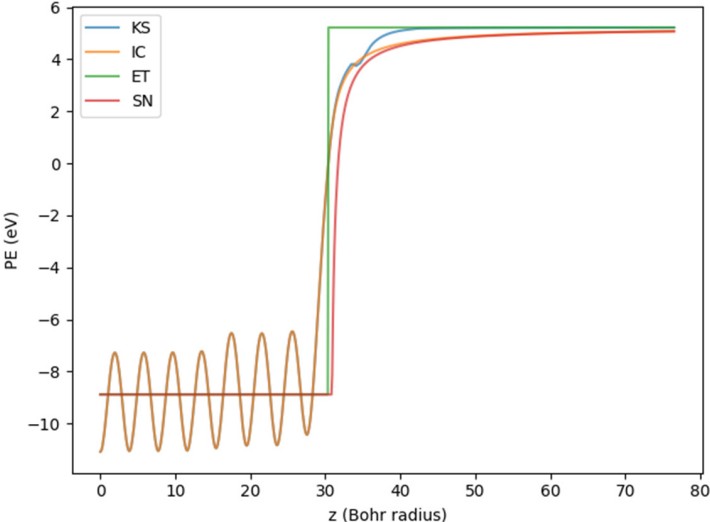

**Figure 4.** The one-dimensional KS, IC, ET, and SN PE profiles at zero field from the center of the slab to the center of the vacuum.

### 2.3. Transmission Coefficient

The WKB approximation assumed the PE barrier to be slowly changing, high and wide on the scale dictated by the electron momentum. This broke down near the metal-vacuum interface, where the PE changed sharply, especially for the ET PE. The assumption no longer holds for a large field, which may reduce the height and width of the PE barrier significantly. To obtain more accurate results and to extend the FN framework to fields above 10 GV/m, we needed to replace the WKB approximation with a more quantitatively accurate method.

To obtain the transmission coefficient, we wanted to solve the one-dimensional TISE for any discretized PE to obtain the full wavefunction. While the TM method is tailored to this problem, there are many numerical techniques to solve general initial value problems for ordinary differential equations. We first performed cubic spline interpolation to obtain a continuous PE. Then, we discretized it on increasingly finer grids to test the convergence of such methods. It turned out that the TM method and the explicit Runge–Kutta method of order 8 (DOP853) [25] had the best accuracy. We examined the ratio of transmission

coefficients from the two methods from 500 MV/m to 80 GV/m (Figure 5). The discrepancy was larger at smaller fields, presumably because of numerical errors inherent in extremely small transmission coefficients. The larger discrepancy for the ET PE might have been caused by its jump discontinuity. Even then, with a sufficiently small step size, the results of the two methods deviated by less than 1% overall. This indicated that both the TM method and DOP853 produced accurate, replicable results for this problem. For computational efficiency, we used the TM method for the following computations.

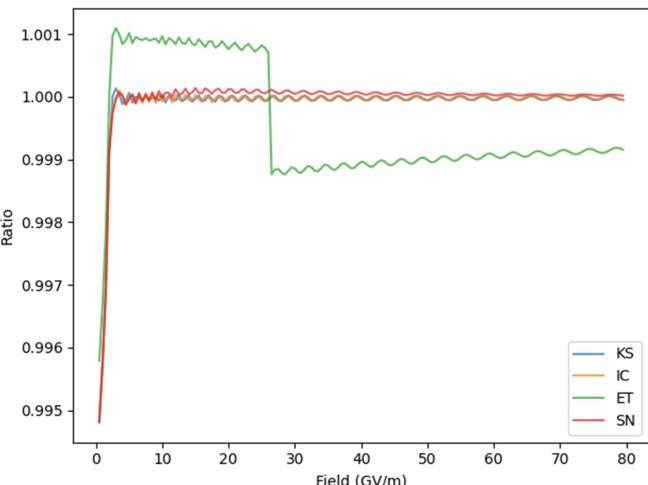

**Figure 5.** Ratio of TM transmission coefficient to that of DOP853 for all PE profiles under consideration.

### 2.4. Emission Current Densities

Once we had the transmission coefficient as a function of electron normal energy, we computed the emission current densities by integrating over the energy-space diagram, assuming a free electron distribution [5]. The general result at zero emitter temperature took the form $J = -z_s \int_{-E_F}^{0} E_n D(E_n) dE_n$, where $E_F$ represented the Fermi energy (Appendix B). According to the free electron assumption, we had $E_F = \frac{\hbar^2}{2m_e} \left(3\pi^2 n\right)^{2/3}$, with $n$ the number density of free electrons in bulk Au. The integral above was then performed numerically.

### 2.5. Summary of Numerical Techniques

To summarize, we first set up a DFT calculation with an Au (100) lattice and selected the appropriate PP and XC functional for this system. The size of the k-point grid and planewave energy cutoff were both determined with convergence tests. This DFT calculation returned a 3D KS PE, so we needed to convert it to 1D for more straightforward solutions. Next, we calculated the KS PE as a function of the field by determining the location of the electrical surface. An IC correction was performed to build a smooth PE profile. To find the transmission coefficient accurately, we chose the TM method. Then, we integrated the transmission coefficient over energy space to obtain the emission current density as a function of the applied field. This procedure is illustrated in Figure 6.

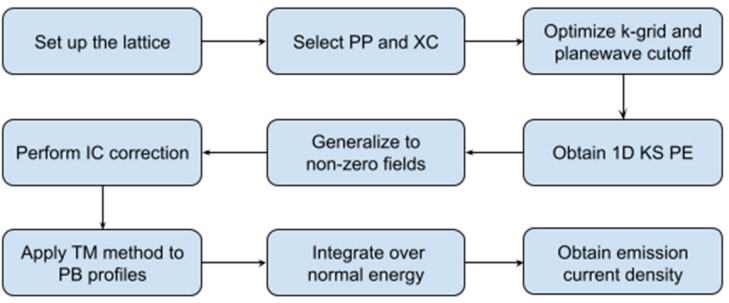

**Figure 6.** Flowchart of numerical procedure.

### 3. Results

*3.1. Transmission Coefficient and Pre-Factor Behavior*

Using the TM method, we calculate the transmission coefficient as a function of the field for all 4 PE profiles, as shown in Figure 7. The reference field, or the field at which the PE profile is reduced to below the Fermi level, is also presented. For the SN PE profile, this reference field is given by $F_R = \frac{4\pi\epsilon_0\phi^2}{e^3} \approx 18.92\,\text{GV/m}$. The reference field does not exist for our KS and IC PE, as their respective surface barriers are never reduced to below the Fermi level. Note that $F_R$ is close to the inflection points of these S-shaped curves of the transmission coefficient. The region beyond $F_R$ is the electron "flyover" regime, where the normal energy of an incident electron is above the PE barrier. Because of quantum mechanical reflection, the transmission coefficient is always smaller than 1. This effect cannot be observed using the WKB approximation, which completely breaks down for fields beyond $F_R$. We also observe that, for all PE profiles, the transmission coefficient slowly approaches unity as the field increases. Quantitatively, the SN transmission coefficient is the greatest, followed by IC, KS, and ET. This quantitative relation reflects that of the PE barrier heights: the SN PE is the lowest, followed by IC, KS, and ET.

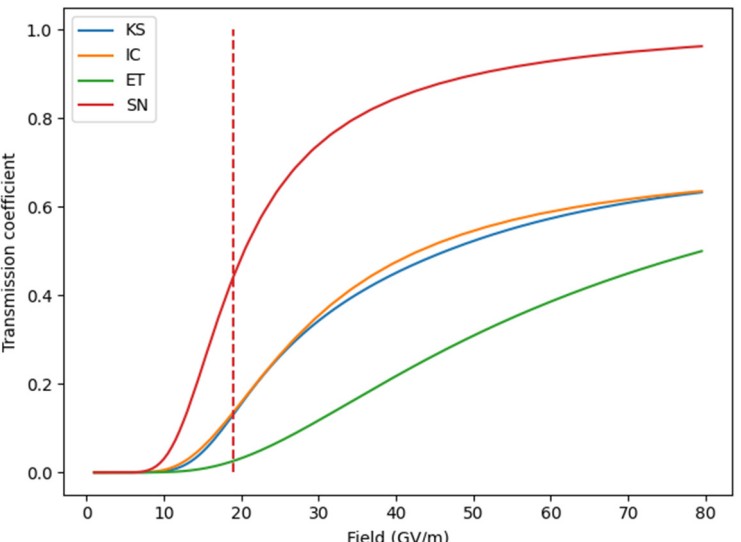

**Figure 7.** Transmission coefficient obtained from the TM method. The dashed red line at 18.92 GV/m represents the reference field $F_R$ for the SN PE.

Forbes argued for the reintroduction of a pre-factor to the WKB transmission coefficient [7]—in the case of a high PE barrier (small fields), the Landau and Lifschitz formula (their Equation (50.12)) provides the relation $D = P_{eff}e^{-G}$ [26]. $P_{eff}$ is an effective pre-factor to the WKB result. For the purpose of comparison with Mayer [27], we present this pre-factor from 1 GV/m up to 80 GV/m and then note the range of applicability of this formula.

Figure 8 presents this pre-factor obtained from the TM method. In the low-field limit, the pre-factor for the ET PE is close to 2, a result consistent with the existing literature [27,28]. The pre-factor for SN is numerically unstable for fields up to 3 GV/m, but for higher fields, it shows a V-shaped curve with the cusp at the reference field $F_R$. This cusp is the result of taking the WKB transmission coefficient to be 1 in the electron "flyover" regime. The V-shaped curve for SN is qualitatively consistent with that of Mayer. Although he presented the pre-factor with increasing electron normal energy, the effect is similar to increasing the field—both methods reduce the height of the PE barrier that an electron sees. Both the KS and IC PEs have comparable pre-factors, which are close to 1 for fields smaller than 10 GV/m but decrease for fields greater than $F_R$. As pointed out by Forbes, the pre-factor is close to unity for "ideally smooth" PE profiles [7]. The low-field behavior of KS and

IC pre-factors is therefore likely due to the "ideally smooth" shape of their PE profiles, in contrast to the jump discontinuity in the ET profile or the steep slope of the SN profile. Due to the "bump" in the KS PE profile, its pre-factor deviates from unity faster than the IC pre-factor. For fields greater than $F_R$, all the pre-factors quickly depart from their low-field behaviors, indicating the breakdown of the Landau and Lifschitz formula at high fields. In summary, our results confirm the necessity of the pre-factor for the ET and SN PEs at fields below $F_R$, while indicating departure from the Landau and Lifschitz formula at higher fields.

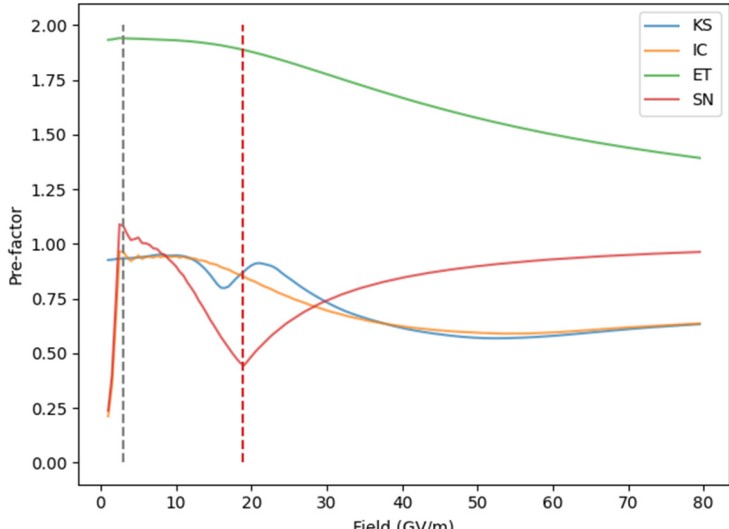

**Figure 8.** Effective pre-factor in the Landau and Lifschitz formula. The dashed gray line at 3 GV/m marks the region of numerical instability to its left. The dashed red line at 18.92 GV/m represents the reference field $F_R$ for the SN PE.

### 3.2. Correction Factor to the First-Order Taylor Expansion

We present an additional correction factor $c$ to the emission current densities, which are traditionally calculated by expanding the Gamow coefficient G to the first order within the integral over the energy space. Our emission current density takes the form $J = cz_s d_F^2 D_F$. Numerical calculations of $d_F$ are presented in Appendix C. For both the first-order Taylor expansion and numerical integration, we use the TM transmission coefficients for consistency.

This correction factor $c$ is close to unity for fields below 10 GV/m but rapidly decreases around $F_R$ (Figure 9). This confirms the general validity of the first-order Taylor expansion for field electron emission, but modifications are necessary in the high-field limit attainable in laser-induced field emission.

### 3.3. Fowler–Nordheim Plot

Finally, we present the emission current density as a function of the field, using the well-known Fowler–Nordheim plot (Figure 10). For an ET PE, the analytical solution on this plot is a straight line with a negative slope. Our results show this behavior for all four PEs, for fields below $F_R$. The slopes of all curves are comparable in this region, so our new PEs (KS and IC) would not account for the experimentally deduced field enhancement factor that tends to increase these slopes [29]. For fields greater than $1/F_R$, all four curves are concave down, indicating that the emission current densities are smaller than expected. This is most likely due to the overprediction by the first-order Taylor expansion in the FN framework (Figure 9). Again, the results for fields above $F_R$ are not easily applicable to ES field emission, but they may be useful as a quasi-static approximation to laser-induced field emission.

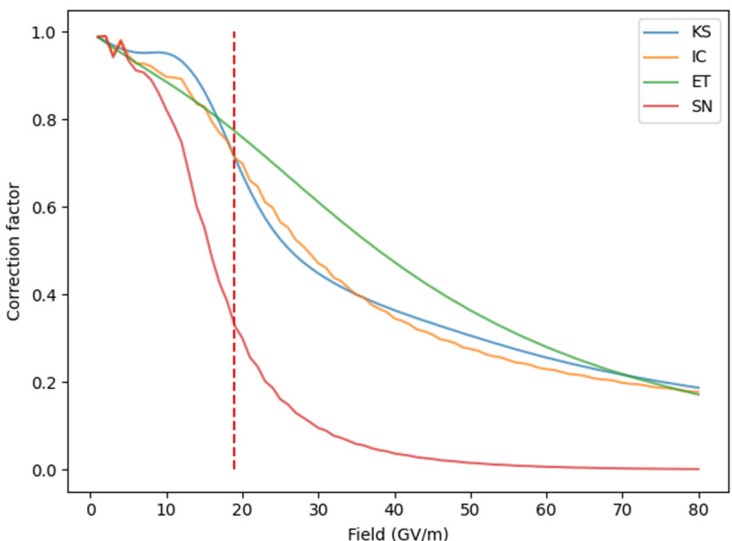

**Figure 9.** Correction factor arising from the first-order Taylor expansion. The dashed red line at 18.92 GV/m represents the reference field $F_R$ for the SN PE.

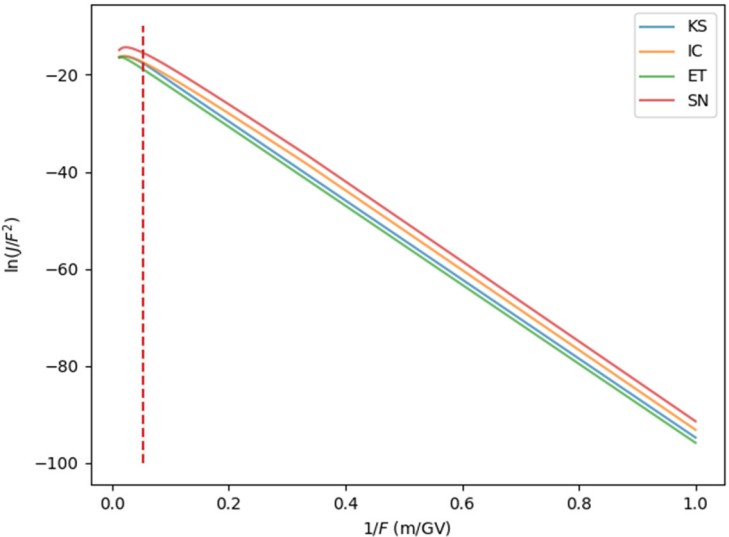

**Figure 10.** Fowler–Nordheim plot of four PEs. $F$ denotes the field and $J$ the emission current density. The horizontal axis is the inverse of the field. The vertical axis is on a logarithmic scale. The dashed red line at $(18.92\,\text{GV/m})^{-1}$ represents the inverse of the reference field $1/F_R$ for the SN PE.

The difference in emission current densities among the four PEs is most pronounced at low fields. The SN PE has the greatest predicted emission current densities, followed by the IC, KS, and ET PEs. This hierarchy is also the order of PE barrier heights from lowest to greatest, as would be expected.

## 4. Discussion

This study is part of the continuing efforts to build accurate numerical models for field electron emission. We used a generalized FN framework for greater fields of up to 80 GV/m by avoiding the WKB approximation and the first-order Taylor expansion of the Gamow coefficient. The correction factors due to these two effects are presented. Except for the effective pre-factor to the ET PE, both correction factors are less than 1 for all PEs in the high field limit. We also obtained the KS and IC PEs from a DFT calculation, and these two PEs produce similar results despite the numerical artifact in the KS PE. The current framework can be employed to study different metal surfaces, including those

with defects and adsorbates, as they can be partly modeled by DFT [30,31]. This study also has implications for laser field emission, which in the high field limit can be studied quasi-statically using instantaneous tunneling currents.

Our numerical method, though, has several limitations. First, the source of the artifact in the one-dimensional KS PE is unclear. It can be examined by optimizing the lattice parameter or using a larger lattice in DFT, both of which are more computationally expensive. Second, the generation of the KS PE at fields above 1 GV/m assumes the electrical surface to be fixed. Although this seems true for fields up to 1 GV/m, this claim needs further examination in the high field limit. Finally, metallic surfaces cannot withstand a static field as large as 80 GV/m, which will necessarily lead to breakdown. Therefore, our results in the high-field limit should be regarded as applicable to transient, ultrafast perturbations only.

Our DFT-based field emission model can be further improved. We have not considered the band structure of a real metallic surface and instead used the free electron model. Neither have we incorporated thermionic effects on the Fermi–Dirac distribution of electrons [28]. Incorporating these two effects would lead to a more complete one-dimensional model of field electron emission. We are well-situated for generalization to a three-dimensional model since the KS PEs are given on a three-dimensional grid. Such a calculation may be helpful in determining the effects of transverse momentum on emission, which could have implications on emitted beam quality in electron sources, such as nanoblades and nanotips. To evaluate this system, one possible solution is to use the three-dimensional TM method to calculate the transmission coefficient [27]. From there, we can then additionally incorporate surface effects, such as defects and adsorbates, into our field electron emission model.

**Author Contributions:** Conceptualization, Y.L. and J.M.; methodology, Y.L.; software, Y.L.; validation, Y.L.; formal analysis, Y.L. and J.M.; investigation, Y.L.; resources, J.M. and J.R.; data curation, Y.L.; writing—original draft preparation, Y.L.; writing—review and editing, J.M. and J.R.; visualization, Y.L.; supervision, J.R.; project administration, J.R.; funding acquisition, J.R. All authors have read and agreed to the published version of the manuscript.

**Funding:** This study was funded by the U.S. National Science Foundation under Award No. PHY-1549132, the Center for Bright Beams.

**Data Availability Statement:** The code can be found at https://github.com/yimingli57/field_electron_emission.git (accessed on 25 November 2023). The DFT results can be obtained from the corresponding author upon reasonable request.

**Conflicts of Interest:** The authors declare no conflict of interest.

## Appendix A

Peak transient fields as large as 80 GV/m have been achieved on gold-coated nanoblades with an ultrafast 800 nm laser via plasmonic enhancement [2,3]. For these values, the Keldysh parameter is $\gamma = \sqrt{\frac{\phi}{2U_p}} \approx 0.23$ [4], where $\phi$ is the work function and $U_p$ the ponderomotive energy. This identifies the tunneling process as dominant when compared to the multiphoton emission process. The transverse motion induced by the laser driver, however, is of moderate concern. Assuming a vacuum planewave driver, the transverse oscillation magnitude induced by the magnetic component of the field is $\beta_0 \approx \frac{U_p \lambda}{4\pi m c^2} \approx 0.0125$ a.u. $\ll 1$ a.u. [32], where 1 a.u. $\approx 52.92$ pm is the atomic unit of length, the Bohr radius. The variable $\lambda$ is the laser wavelength, $m$ the electron mass, and $c$ the speed of light in vacuum. This puts us squarely within the "tunnel oasis", permitting the use of a quasi-static tunneling model.

Even so, considering that the fields are achieved with plasmonic enhancement, the ratio of the magnetic field strength at a flat metal–vacuum boundary compared to that for a planewave in a vacuum is $\frac{1+\epsilon}{\epsilon^2}$ for $\epsilon$, the real component of the relative permittivity [33]. At

800 nm the permittivity's magnitude is approximately $|\epsilon| \approx 25$ [34], and thus, the magnetic field would have even less an effect than would be assumed by the above equation for $\beta_0$.

On the other hand, there are transverse electric fields (transverse to the surface normal, along the propagation direction) in surface plasmons as well. The transverse field strength compared to the longitudinal field strength is $\left|\frac{E_x}{E_0}\right| = \frac{1}{\sqrt{|\epsilon|}}$. The transverse ponderomotive amplitude due to this field is $a_{p,\perp} = \frac{e|E_x|}{m\omega^2} = \frac{eE_0\lambda^2}{4\pi^2\sqrt{|\epsilon|}mc^2}$. With the same values as above, we get a transverse oscillation magnitude of $a_{p,\perp} \approx 10$ a.u. It should be noted that the motion here is along the ridge of the nanoblade under consideration, which is smooth and thus transversely uniform. While this motion is large on the atomic scale, it is negligible on the next length scale—the apex radius of curvature, with a minimal estimate of 20 nm $\approx 378$ a.u. [3]. Thus, the tunneling process may again be treated in one dimension and quasi-statically without the consideration of magnetic or transverse electric effects.

**Appendix B**

According to the free electron assumption, the electrons are evenly distributed in total energy and normal energy below the Fermi level. The emission current density takes the form $J = z_s \iint D\, dK_p dE$, where $K_p$ is the surface-parallel kinetic energy and $E$ the electron total energy [5]. In our calculation, the transmission coefficient is only a function of electron normal energy $E_n$, so we use the change of variables to convert the integral. The complete integral can be written as follows:

$$J = z_s \int_{-E_F}^{0} \int_{0}^{E_F+E} D(E - K_p)dK_p dE$$

By conservation of energy, $E_n = E - K_p$, so we can change the inner integral from $K_p$ to $E_n$:

$$J = z_s \int_{-E_F}^{0} \int_{-E_F}^{E} D(E_n)dE_n dE$$

Now, we reverse the order of integration:

$$J = z_s \int_{-E_F}^{0} \int_{E_n}^{0} D(E_n)dE dE_n$$

The inner integral over $E$ no longer explicitly depends on $E$, so it can be performed directly. The final result is as follows:

$$J = -z_s \int_{-E_F}^{0} E_n D(E_n)dE_n$$

**Appendix C**

Without the WKB approximation, the inverse of the decay width is defined as $d_F^{-1} = \frac{\partial}{\partial E_n}\ln(D(E_n))|_F$, evaluated at the Fermi level $E_n = E_F$. $D(E_n)$ is calculated numerically using the TM method. This expression is then differentiated numerically using the central-difference method. The initial step size is 0.1 eV, and it is halved after each iteration until the result from the current iteration converges to within 0.01% of the result from the previous iteration. The results of numerical differentiation are presented in Figure A1. Note that the decay width for the SN PE blows up, another suggestion of the breakdown of the traditional FN framework for fields greater than $F_R$.

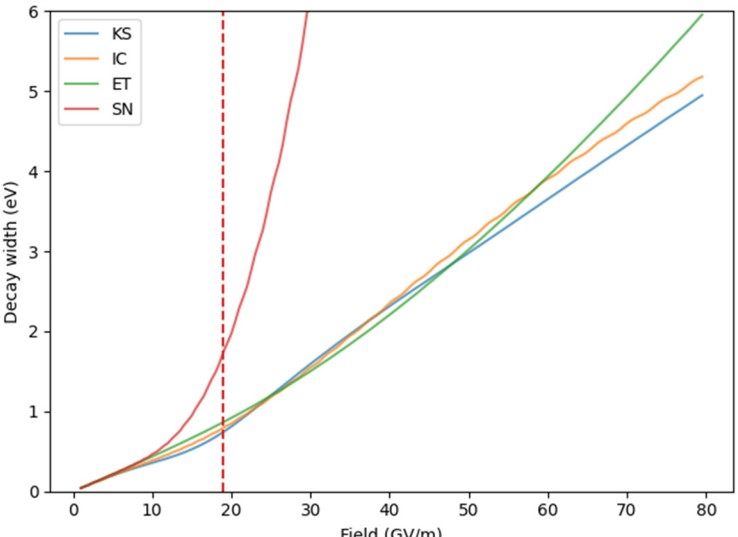

**Figure A1.** Decay width as a function of the applied field for four PEs. The dashed red line at 18.92 GV/m represents the reference field $F_R$ for the SN PE.

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
