# Peer review of "Modeling Field Electron Emission from a Flat Au (100) Surface with Density-Functional Theory"

_instruments, doi:10.3390/instruments7040047_

Round 1
Reviewer 1 Report
Comments and Suggestions for Authors
See separate file

Author Response
Dear reviewer,
We are very grateful for your detailed, well-researched and well-articulated feedback on our manuscript. We find these suggestions very illuminating and constructive. We have carefully considered all the points you bring up and modified our manuscript accordingly.
- We have read and incorporated references [R1] and [R2] into our manuscript when presenting numerical techniques to find the transmission coefficient. We compare our pre-factor with the results in [R2] and note similarities in our approaches.
- After thoroughly reading [R3], we follow Reiss’ calculation to show that we are in the “tunneling oasis” for the laser parameters considered. We explain why the effects of the magnetic field and transverse electric field would not prohibit us from applying the quasi-static approximation to laser-induced field emission. We therefore think the relation between ES field emission and laser-induced field emission is still justified.
- We define 80 GV/m as the “peak surface-enhanced electric field” in a laser pulse. We further state that this limit is only applicable to the quasi-static approximation of laser-induced field emission, and results above 10 GV/m typically should not be applied to ES field emission.
- In Section 3.1, we directly plot the transmission coefficient over the ES field, and we include the position of the reference field. We keep the plot on the pre-factor, but we compare our results with those found by Mayer. Based on this plot, we also explain the breakdown of the LL formalism that gives this pre-factor.
- In Section 3.2 and 3.3, we indicate the reference field on both plots and explain the regions that are accessible to ES field emission experiments.
- We correct our use of terminology following the reviewer’s suggestion. We replace the term “electric field” by “field” in the context of ES field emission and define its meaning in our introduction.
Please feel free to provide any remark on our revised manuscript. We are looking forward to hearing from you!
Reviewer 2 Report
Comments and Suggestions for Authors
The work is interesting and well presented, well detailed in the methods with their advantages. The final discussions are well argued with respect to the limits of the work and possible future developments.
I have just a few comments. Once reviewed, I think that this work is suitable for publication, and it may stimulate new numerical developments on this topic.

Author Response
Dear reviewer,
Thank you for taking your time to review our manuscript and providing valuable feedback. We have considered all of your points and modified our manuscript accordingly.
- In the revised introduction, we explain the applicability of the quasi-static approximation to laser-induced field emission in greater detail. We also presents previous high-precision methods for finding the transmission coefficient.
- We add a figure to explain the advantage of the PBE exchange-correlation functional.
- We add a diagram showing the atomic positions in our Au (100) system.
- We include a plot showing the average induced charge density along the longitudinal direction and the centroid location.
- In the added Section 2.5, we present a flowchart to summarize the numerical procedure presented above.
We would like to thank you again for your constructive feedback!
Round 2
Reviewer 1 Report
Comments and Suggestions for Authors
None